# Energy-Efficient Harmonic Transponder Based on On-Off Keying Modulation for Both Identification and Sensing

**DOI:** 10.3390/s22020620

**Published:** 2022-01-14

**Authors:** Valentina Palazzi, Luca Roselli, Manos M. Tentzeris, Paolo Mezzanotte, Federico Alimenti

**Affiliations:** 1Department of Engineering, University of Perugia, 06125 Perugia, Italy; luca.roselli@unipg.it (L.R.); paolo.mezzanotte@unipg.it (P.M.); federico.alimenti@unipg.it (F.A.); 2School of Electrical and Computer Electronics, Georgia Institute of Technology, Atlanta, GA 30332, USA; etentze@ece.gatech.edu

**Keywords:** frequency doubler, harmonic radar, harmonic transponder, internet of things

## Abstract

This paper presents a novel passive Schottky-diode frequency doubler equipped with an on-off keying (OOK) modulation port to be used in harmonic transponders for both identification and sensing applications. The amplitude modulation of the second-harmonic output signal is achieved by driving a low-frequency MOSFET, which modifies the dc impedance termination of the doubler. Since the modulation signal is applied to the gate port of the transistor, no static current is drained. A proof-of-concept prototype was manufactured and tested, operating at 1.04 GHz. An on/off ratio of 23 dB was observed in the conversion loss of the doubler for an available input power of −10 dBm. The modulation port of the circuit was excited with a square wave ($f_{m}$ up to 15 MHz), and the measured sidebands in the spectrum featured a good agreement with the theory. Then, the doubler was connected to a harmonic antenna system and tested in a wireless experiment for $f_{m}$ up to 1 MHz, showing an excellent performance. Finally, an experiment was conducted where the output signal of the doubler was modulated by a reed switch used to measure the rotational speed of an electrical motor. This work opens the door to a new class of frequency doublers, suitable for ultra low-power harmonic transponders for identification and sensing applications.

## 1. Introduction

Harmonic transponders have gained a growing popularity in Internet of Things applications [1,2,3,4]. Despite their drawbacks, which include the high conversion loss in the harmonic generation and the higher path loss experienced by the back-scattered higher-order harmonic components with respect to the fundamental tone, they have key features for many application scenarios. These tags can operate without any dc power supply, which means that they do not require a periodic battery replacement. Moreover, thanks to their capability to use different frequencies for the uplink and the downlink, they are robust to clutter, and they are not subject to the self-jamming problems of traditional radiofrequency identification (RFID) systems [5,6]. So, they can be placed in harsh environments and in remote areas. These features justify their application for insect-tracking [7,8,9], condition monitoring [10,11,12,13], and search-and-rescue applications [14,15].

Harmonic transponders are simply based on nonlinear elements (usually zero-bias diodes), which operate as frequency multipliers, connected to antenna systems [16,17]. Similar to typical RFIDs, these transponders are interrogated by a sinusoidal signal at a frequency $f_{0}$. The acquired signal is distorted by the non linearity of the tag, which generates harmonic frequencies. The generated *n*-th order harmonic is then back-scattered towards the receiver tuned to $n\;f_{0}$.

Although these transponders can straightforwardly provide one-bit information (i.e., communicate their presence/absence), there are applications where more complex information, such as a tag identifier or sensor data, is needed. Some attempts to encode additional information in harmonic tags have been already reported in the literature. Most harmonic transponders encode sensor information statically in the power of the back-scattered second harmonic [11,12]. However, the received power also depends on the channel and on the tag-to-reader distance, which makes information recovery prone to errors. In [18] a harmonic sensing platform was presented, where the sensor information was encoded in the phase difference between signals back-scattered in two orthogonal polarizations. This solution, though, caused an increase in the number of components in the tag and in the reader.

In [13,19,20,21] identification information was encoded in the amplitude modulation of the second harmonic. In [19] the modulation was achieved by varying the bias point of varactors along a nonlinear transmission line using a digital modulation unit. In [20] an implantable system was proposed for wireless neurorecording, based on a varactor pair, which mixed the fundamental RF carrier with the low-frequency and low-voltage signals represented by the neuropotentials. In [13] a low-voltage oscillator was used to periodically change the dc bias point of a Schottky diode. In [21] a radiofrequency (RF) switch was placed between the $f_{0}$ antenna and the doubler to periodically vary the input matching of the doubler. In the above papers, the modulation was obtained either by varying the bias point of the nonlinear element or by varying the impedance in the RF path, and high-frequency modulation circuitry was needed.

State-of-the-art frequency doublers and quadruplers with modulation capability have also been reported at mm-wave and sub-THz frequency ranges [22,23,24]. These frequency multipliers were active, and the signal modulation was achieved using transistors either acting as RF switches or switching on and off the bias of the circuit itself (amplifier or multiplier stages).

In this work a novel Schottky-diode frequency doubler (n=2) with OOK modulation capability is presented, which relies on a low-frequency MOSFET switch placed in the output matching network of the doubler. The novelty of the proposed circuit lies in the fact that the switch controls the impedance along the path of the self-generated dc signal component flowing through the doubler, while it does not affect the RF signal components. This way, the OOK modulation can be enabled even by switches characterized by large parasitics, opening the door to innovative modulation strategies. First, the theoretical analysis of the circuit is performed; then, the obtained experimental results are discussed, and finally conclusions are drawn.

## 2. Theory

This paper presents a passive frequency doubler equipped with a modulation port for RFID applications. The block diagram of the proposed system is shown in Figure 1. The system consists of a reader and a tag. The reader transmits a sinusoidal signal with a frequency f0. This signal is captured by the input antenna of the tag and conveyed to the doubler (“$x_{in}$” in Figure 1). The doubler both converts the signal to the second harmonic and perform an OOK modulation of the signal. The obtained output signal (“$x_{out}$” in Figure 1) is finally back-scattered toward the reader by means of the output antenna of the tag, and it is detected by the receiver. The information can be retrieved through downconversion and demodulation of the received signal.

Figure 2 shows the schematic diagram of the doubler. The circuit is based on a series-connected low-barrier Schottky diode (model HSMS-2850) [25]. Schottky diodes, which are commonly used in frequency multipliers [26], are chosen for their capability to operate without any bias signals and for small available RF input powers. Two quarter-wave short-circuited and open-circuited stubs, connected in parallel at the two sides of the diode, operate as harmonic filters. The short-circuited stub connected at the anode of the diode behaves as an open circuit for the $f_{0}$ tone and as a short circuit for the $2f_{0}$ tone. On the other hand, the open-circuited stub connected at the cathode of the diode behaves as a short circuit for the $f_{0}$ tone and as an open circuit for the $2f_{0}$ tone. This way, the $f_{0}$ tone at the input port can flow through the diode. The $2f_{0}$ tone generated by the diode, instead, can flow through the output load, while the dynamic ground at the diode anode prevents this component from flowing toward the source.

The input matching network, consisting of a distributed tapped impedance transformer, is used to match the large-signal input impedance of the circuit to the source impedance at the specific designed input power and frequency of the doubler. The output matching network, consisting of an LC network ($L_{m}$ and $C_{m}$ in Figure 2), is used to transform the load impedance of the doubler into the optimum impedance that minimizes the conversion loss of the circuit, while behaving as a bias tee (i.e., the self-generated dc component of the diode has a return through the inductor $L_{m}$).

The main difference with respect to [25] is that a modulation mechanism is added to the present circuit. The modulation is achieved by changing the dc impedance termination of the diode, as it significantly affects the diode self-biasing.

To demonstrate this concept, the channel conductance of an NMOS transistor in triode region is used. Such conductance is controlled by biasing the gate with no static current consumption, which is ideal for ultra-low power applications. The transistor is connected between the inductance $L_{m}$ and ground (see Figure 2), while the bypass capacitor $C_{b}$ ensures the proper RF operation of the output matching network ($L_{m}$, $C_{m}$). When the gate voltage is above the threshold, the channel is formed and the Schottky diode is zero biased. In this case (“on” condition) the frequency doubler operates as in [25] with the minimum conversion loss. On the other hand, when the gate voltage is below the threshold or zero, the channel is not formed, the diode cathode is open at dc, and the output capacitances $C_{b}$ and $C_{m}$ are quickly charged, developing a positive voltage. As a consequence, the diode is reverse biased, and the conversion loss increases significantly (“off” condition). In this way, the diode self-biasing is used to inhibit the circuit operation in the “off” state.

The doubler equivalent circuits for different harmonic components is shown in Figure 3. To simplify the analysis, the diode parasitics and the nonlinear diode junction capacitance are neglected. Harmonic components higher than the second harmonic are also neglected. $R_{s}$ is the equivalent resistance seen by the diode at $f_{0}$ looking toward the source; $R_{l}$ is the equivalent resistance seen by the diode at $2f_{0}$ looking toward the load. The voltage across the diode $v_{D}\left(t\right)$ can be approximated as follows: (1)$$v_{D}\left(t\right)=v_{0}+v_{1}\cos\left(\omega_{0}t\right)+v_{2}\cos\left(2\omega_{0}t\right),$$
where $v_{k}$ (k=0,1,2) are the voltage coefficients associated with the dc, $f_{0}$, and $2f_{0}$ components, respectively. They are real coefficients, since the circuit is resistive, based on the above mentioned simplifying hypothesis.

The current flowing through the diode $i_{D}\left(t\right)$ can be determined by substituting (Equation 1) in the exponential IV model: (2)$$i_{D}\left(t\right)=I_{s}\left[e^{\frac{v_{D}\left(t\right)}{nV_{T}}}-1\right]=I_{s}\left[e^{\frac{v_{0}}{nV_{T}}}e^{\frac{v_{1}}{nV_{T}}\cos\left(\omega_{0}t\right)}e^{\frac{v_{2}}{nV_{T}}\cos\left(2\omega_{0}t\right)}-1\right]$$
where $I_{s}$ is the reverse saturation current, *n* is the ideality factor, and $V_{T}$ is the thermal voltage.

For $P_{in}=-10$ dBm, $v_{1}$ cannot be considered a small value ($v_{1}$ in the range 0.2–0.5 V according to the performed Harmonic-Balance simulations); therefore we need to apply a large-signal analysis. According to [27], the exponential term associated with the fundamental tone can be expanded in the Fourier series using the modified Bessel functions of the first kind, $I_{n}\left(x\right)$, as follows: (3)$$e^{\frac{v_{1}}{nV_{T}}\cos\left(\omega_{0}t\right)}=I_{0}\left(\frac{v_{1}}{nV_{T}}\right)+2I_{1}\left(\frac{v_{1}}{nV_{T}}\right)\cos\left(\omega_{0}t\right)+2I_{2}\left(\frac{v_{1}}{nV_{T}}\right)\cos\left(2\omega_{0}t\right)+\ldots$$
where the expansion is truncated at the second order.

On the other hand, a small-signal analysis can be applied to expand the exponential term associated with the second harmonic, since $v_{2}$ is small ($v_{2}$ below 40 mV according to the performed Harmonic-Balance simulations); therefore, we can use the Taylor series: (4)$$e^{\frac{v_{2}}{nV_{T}}\cos\left(2\omega_{0}t\right)}=1+\frac{v_{2}}{nV_{T}}\cos\left(2\omega_{0}t\right)+\ldots$$
where the expansion is truncated at the first order.

Substituting (Equation 3) and (Equation 4) into (Equation 2), we obtain the current amplitudes $i_{k}$ (k=0,1,2) of the diode current $i_{D}\left(t\right)$, associated with dc, $f_{0}$, and $2f_{0}$:(5)$$\begin{array}{c} i_{D}\left(t\right)=i_{0}+i_{1}\cos\left(\omega_{0}t\right)+i_{2}\cos\left(2\omega_{0}t\right)=I_{s}\left[I_{0}\left(\frac{v_{1}}{nV_{T}}\right)e^{\frac{v_{0}}{nV_{T}}}-1\right]+\\ +2I_{s}I_{1}\left(\frac{v_{1}}{nV_{T}}\right)e^{\frac{v_{0}}{nV_{T}}}\cos\left(\omega_{0}t\right)+2I_{s}\left[I_{2}\left(\frac{v_{1}}{nV_{T}}\right)+\frac{v_{2}}{2nV_{T}}I_{0}\left(\frac{v_{1}}{nV_{T}}\right)\right]e^{\frac{v_{0}}{nV_{T}}}\cos\left(2\omega_{0}t\right) \end{array}$$
As shown by Figure 3a, when the switch is closed the dc voltage component across the diode $v_{0}$ is zero. Equation (Equation 5) simplifies as follows:(6)$$\begin{array}{c} i_{D}\left(t\right)=I_{s}\left[I_{0}\left(\frac{v_{1}}{nV_{T}}\right)-1\right]+2I_{s}I_{1}\left(\frac{v_{1}}{nV_{T}}\right)\cos\left(\omega_{0}t\right)+\\ +2I_{s}\left[I_{2}\left(\frac{v_{1}}{nV_{T}}\right)+\frac{v_{2}}{2nV_{T}}I_{0}\left(\frac{v_{1}}{nV_{T}}\right)\right]\cos\left(2\omega_{0}t\right). \end{array}$$

When the switch is open, instead, the diode dc current $i_{0}$ is zero. This implies that
(7)$$e^{\frac{v_{0}}{nV_{T}}}=\frac{1}{I_{0}\left(\frac{v_{1}}{nV_{T}}\right)},$$
or, equivalently, that: (8)$$v_{0}=-nV_{T}\ln\left[I_{0}\left(\frac{v_{1}}{nV_{T}}\right)\right].$$

Since $I_{0}\left(x\right)$ is a monotonically increasing function of $x=v_{1}/\left(nV_{T}\right)$, greater than 1, $v_{0}$ is negative. Therefore, the diode is reverse biased.

For x≫1 and for any integer order *n*, $I_{n}\left(x\right)$ can be approximated as [28]: (9)$$I_{n}\left(x\right)\approx \frac{e^{x}}{\sqrt{2\pi x}}.$$

This approximation can be applied to the present case, since $v_{1}$ is in the range 0.2–0.5 V, while n=1.06 and $V_{T}\approx 25$ mV at room temperature. The approximation in (Equation 9) can be used in (Equation 8) to calculate $v_{0}$: (10)$$v_{0}\approx -v_{1}+\frac{nV_{T}}{2}\ln[2\pi \frac{v_{1}}{nV_{T}}].$$

Substituting (Equation 7) into (Equation 5) we obtain the diode current:(11)$$i_{D}\left(t\right)=2I_{s}\frac{I_{1}\left(\frac{v_{1}}{nV_{T}}\right)}{I_{0}\left(\frac{v_{1}}{nV_{T}}\right)}\cos\left(\omega_{0}t\right)+2I_{s}\left[\frac{I_{2}\left(\frac{v_{1}}{nV_{T}}\right)}{I_{0}\left(\frac{v_{1}}{nV_{T}}\right)}+\frac{v_{2}}{2nV_{T}}\right]\cos\left(2\omega_{0}t\right).$$

According to (Equation 9), for x≫1
(12)$$I_{n}\left(x\right)/I_{0}\left(x\right)\to 1.$$

Therefore, (Equation 11) further simplifies as follows: (13)$$i_{D}\left(t\right)=2I_{s}\cos\left(\omega_{0}t\right)+2I_{s}\left[1+\frac{v_{2}}{2nV_{T}}\right]\cos\left(2\omega_{0}t\right).$$

The voltage coefficient of the fundamental frequency component $v_{1}$ can be derived from the equivalent circuit in Figure 3b: (14)$$v_{1}=v_{s}-R_{s}i_{1}=v_{s}-\frac{R_{s}}{r_{D}}v_{1},$$
where $r_{D}$ is the large-signal diode impedance at the fundamental frequency. Therefore, $v_{1}$ can be expressed as
(15)$$v_{1}=\frac{v_{s}}{1+\frac{R_{s}}{r_{D}}}.$$

The input matching network is designed so that $r_{D}=R_{s}$ for a specific design input power (equal to −10 dBm in our case) when the switch is closed. Therefore, for such a power (matching condition): (16)$$v_{1}\approx \frac{v_{s}}{2}.$$

The voltage amplitude when the switch is open, instead, can be determined by substituting $i_{1}$ from (Equation 13) into (Equation 14): (17)$$v_{1}=v_{s}-2R_{s}I_{s}\approx v_{s}.$$

The latter approximation holds since $I_{s}$ is significantly smaller than $v_{s}$.

Finally, the voltage amplitude of the second harmonic component can be derived from the equivalent circuit in Figure 3c: (18)$$v_{2}=-R_{l}i_{2}.$$

When the switch is closed, $i_{2}$ can be obtained from (Equation 6): (19)$$v_{2}=-2R_{l}I_{s}\left[I_{2}\left(\frac{v_{1}}{nV_{T}}\right)+\frac{v_{2}}{2nV_{T}}I_{0}\left(\frac{v_{1}}{nV_{T}}\right)\right].$$

Solving (Equation 19) for $v_{2}$ and applying the large argument approximations for the modified Bessel functions in (Equation 9) and (Equation 12) (asymptotic case), we obtain
(20)$$v_{2}\approx -2nV_{T}.$$

A similar procedure can be followed to determine $v_{2}$ when the switch is open: (21)$$v_{2}\approx -\frac{2R_{l}I_{s}}{1+\frac{R_{l}I_{s}}{nV_{T}}}.$$

In the proposed circuit, $I_{s}=3\;\mu$A, $R_{s}=280\;\Omega$, and $R_{l}=130\;\Omega$. Therefore, at $P_{in}=-10$ dBm, $v_{s}=473$ mV. This means that when the switch is open $v_{0}=-409$ mV, $v_{1}=471$ mV, and $v_{2}=-0.8$ mV. When the switch is closed, instead, $v_{1}=236$ mV and $v_{2}=-53$ mV. Although this analysis is simplified, it shows that the second harmonic voltage when the switch is closed is significantly larger than the voltage when the switch is open. This is the very principle used to perform the OOK modulation. The obtained values compare well with the results obtained from the Harmonic-Balance simulation of the circuit in Figure 2. The comparison between the simulated diode voltage and the model is shown in Figure 4.

On a first approximation, to obtain an estimate of the output spectrum we assume that the output signal is null when the MOSFET is off. We assume that the MOSFET is on with a duty cycle of 50% (digitally, it corresponds to transmitting a sequence of alternate zeroes and ones). The output signal $x_{out}$ is obtained by multiplying the second harmonic generated by the diode by a square wave:(22)$$x_{out}\left(t\right)=A_{on}\;\cos\left(2\;\omega_{0}t\right)\;\left[\frac{1}{2}+\frac{2}{\pi }\;\cos\left(\omega_{m}\;t\right)-\frac{2}{3\pi }\;\cos\left(3\;\omega_{m}\;t\right)+\ldots \right],$$
where $A_{on}$ corresponds to the amplitude of the output signal in the “on” state, $\omega_{0}=2\pi f_{0}$, and $\omega_{m}=2\pi f_{m}$ is the modulation frequency. The spectrum of the output signal corresponds to the convolution of the Fourier transform of the two signals:(23)$$\begin{array}{c} |X_{out}\left(f\right)|=\frac{A_{on}}{2}\delta \left(f-2f_{0}\right)+\frac{A_{on}}{\pi }\delta \left(f-2f_{0}\pm f_{m}\right)+\\ +\frac{A_{on}}{3\pi }\delta \left(f-2f_{0}\pm 3f_{m}\right)+\ldots \end{array}$$

Based on (Equation 23), ideally the output signal at $2f_{0}$ is 6 dB below that of the doubler in the “on” state. Additionally, the ratio of the magnitude of the first sideband to the carrier is equal to 2/π, which means that the first sidebands are about 4 dB lower than the carrier.

The conversion loss *CL* of the circuit is defined as
(24)$$CL\left(on/off\right)=\frac{P_{in}^{f0}}{P_{out}^{2f0}\left(on/off\right)},$$
where $P_{in}^{f0}$ is the available input power at $f_{0}$, and $P_{out}^{2f0}$ is the output power at $2f_{0}$ delivered to the load in the on and off states, respectively. Note that due to the fact that the signal is directly applied to the gate no static current is required to activate the transistor. Additionally, since the MOSFET impacts only the dc signal component, low-frequency transistors can be used. The modulation signal can encode a digital identification number, or can be a periodic signal generated by a transducer (such as, for instance, a piezoelectric transducer for vibration sensing).

## 3. Experimental Results

The proposed circuit was designed in microstrip technology on an FR4 substrate (h=0.8 mm, $\epsilon_{r}=4.7$, and tan δ=0.011). All circuit components were implemented as distributed elements except for the diode, the MOSFET, and the capacitor $C_{m}$. The bypass capacitor $C_{b}$, in parallel with the transistor, can be omitted, as it is absorbed into the parasitic drain-source (output) capacitance of the BSS123. For demonstration purposes, the fundamental frequency $f_{0}$ was set to 1.04 GHz, the design available input power $P_{in}^{f0}$ was set to −10 dBm, and the source and load impedances were equal to 50Ω. This value of $f_{0}$ was chosen for demonstration purposes to test the circuit in the low-GHz range. Figure 5 shows a photo of the complete prototype. The design was carried out with the help of the Advanced Design System suite adopting a co-simulation approach: the distributed parts of the circuit were electromagnetically simulated with Momentum and then interfaced with the circuit models of the lumped components in a Harmonic-Balance simulation. The lines were folded to obtain a compact layout.

To test the circuit in static condition, an RF signal generator was connected to the RF input port of the doubler, while a spectrum analyzer was connected to the output port. A power supply was used to switch the MOSFET on and off.

The output spectrum of the doubler at $P_{in}=-10$ dBm and $f_{0}=1.04$ GHz with the MOSFET switched on and off, respectively, is shown in Figure 6a. The supply voltage for the “on” state was set to 3 V. The measured RF output power when the MOSFET was off was attenuated by 23 dB with respect to the output power when the MOSFET was on. Figure 6b shows the conversion loss of the doubler versus its available input power at the design frequency of 1.04 GHz, when the MOSFET was switched on (square symbols) and off (circle symbols). The RF input power varied from −25 dBm to 5 dBm. The results of the simulations are included for comparison. The small discrepancies between simulations and measurements are due to inaccuracies in the component models and due to manufacturing tolerances. At the design power of −10 dBm, the measured *CL* was 15.1 dB when the MOSFET was on and equal to 38.2 dB when the MOSFET was off. It is worth noticing that this was achieved exclusively through the variation of the dc termination. The on/off ratio of the conversion loss was almost constant for larger RF input powers, while it decreased for lower powers. This is due to the fact that the self-generated dc voltage counter-biasing the diode is very small in the linear region. Nevertheless, an on/off ratio of about 15 dB was still observed for $P_{in}=-20$ dBm.

In Figure 6c the conversion loss versus frequency at the design power level of −10 dBm is shown. The dynamic range between the “on” and “off” states was above 23 dB throughout the whole band from 940 MHz to 1.14 GHz.

The doubler was then tested in a dynamic condition. A square wave, with a peak-to-peak voltage amplitude of 3 V was applied to the gate of the MOSFET with a waveform generator. The modulation frequency was varied from 500 kHz to 15 MHz (the maximum frequency allowed by the waveform generator available in our laboratory, model HP 33120A). A buffer (model SN74AUP1G34 from TI) was inserted in the series between the waveform generator and the gate port of the MOSFET to improve the quality of the applied voltage square wave. The measured output spectrum is shown in Figure 7. The maximum modulation frequency at which the present circuit was tested was noticeably higher than the maximum modulation frequency used by standard UHF RFIDs [29], which was equal to 128 kbps, demonstrating the capability of the circuit to support high data rates.

Figure 8 shows a comparison between the measured output frequency spectrum at $f_{m}=1$ MHz and $f_{m}=15$ MHz for $P_{in}$ equal to −10 and −20 dBm and the spectrum values predicted by (Equation 23). The first sidebands at ±1 MHz were 5.2 dB and 5 dB below the carrier for $P_{in}=-10$ dBm and $P_{in}=-20$ dBm, respectively, while the first sidebands at ±15 MHz were 6.35 dB and 5.25 dB below the carrier. These values were larger than the 4 dB predicted by the developed simplified theory. Additionally, faint spectral components were noticed at $2f_{0}\pm 2f_{m}$, $2f_{0}\pm 4f_{m}$,…, which were not predicted by (Equation 23), and the discrepancy between measurements and theory increased as the order of the sidebands increased. These differences are due to the fact that (Equation 23) represents the output spectrum obtained multiplying the $2f_{0}$ tone by an ideal square wave. It did not consider that the applied voltage square wave is affected by transients and small asymmetries, that a small output power is generated by the doubler when the MOSFET is off, and that the MOSFET switch has its own transfer function. Nevertheless, the amplitude modulation was still well recognizable in both cases.

The dynamic power consumption ($P_{c}$) of the modulator was measured as the additional power consumption of a digital buffer used to feed the modulator. The buffer (model SN74AUP1G34 from TI) was inserted in the series between the waveform generator and the gate port of the MOSFET. The power consumption of the modulator was obtained as the difference between the dc power consumption of the buffer connected and disconnected from the waveform generator. $P_{c}$, shown in Figure 9, increased linearly with $f_{m}$, and it was equal to 442 μW for $f_{m}=1$ MHz. This corresponds to charging and discharging a capacitor of about 50 pF.

Finally, the doubler was tested in a wireless harmonic transponder (see Figure 5c). The input and output ports were connected to patch antennas tuned to $f_{0}$ and $2f_{0}$, respectively. A transmitted RF power of 16 dBm EIRP was used to interrogate the tag at $f_{0}$, while the receiver consisted of a spectrum analyzer connected to a 10 dB-gain antenna. The tag-to-reader distance was 50 cm (corresponding to $P_{in}=-10$ dBm).

Although the $2f_{0}$ antenna connected to the spectrum analyzer was in near field (the $f_{0}$ reader antenna was 25-cm long, while the $2f_{0}$ reader antenna was 23-cm long, which corresponded to a far-field distance of 44 cm for the $f_{0}$ antenna and 73 cm for the $2f_{0}$ antenna), this setup was sufficient to proof the concept, which consisted of verifying the shape of the spectrum produced by the proposed transducer when interrogated in a wireless transponder. The measured received spectrum for different modulation frequencies is reported in Figure 10. All spectra had a similar shape, with the first sidebands about 5 dB below the carrier for $f_{m}$ in the range 100 kHz–1 MHz. Higher modulation frequencies could not be tested, due to the limited operating bandwidth of the $2f_{0}$ antenna used for the tag. In Figure 8, we observed that the doubler operated correctly for $P_{in}$ higher than −20 dBm. With the proposed wireless experimental setup, this corresponded to a maximum tag-to-reader distance of 1.6 m.

## 4. Sensor Application

In this paper, we presented a frequency doubler, where an OOK modulation was achieved by changing the dc impedance termination of a Schottky diode. This method is innovative with respect to state-of-the-art (SoA) solutions based on bias signals: indeed, the dc impedance termination of the diode can be modified even with components characterized by large parasitics and with methods which do not require any electrical signals, leading to fully passive solutions and innovative sensing approaches. As an example, in Figure 11, the dc impedance termination of the doubler was modified with a reed switch (the schematic of the circuit is shown in Figure 12). The MOSFET was removed and the two terminals of the reed switch were soldered in parallel with the bypass capacitor ($C_{b}=100\;$pF).

This electromechanical switch, actuated by a magnet, featured high parasitic inductances and capacitances, which made it unusable at RF. However, in the proposed solution, the switch impacted only the dc signal component self-generated by the diode, so it could be profitably used, as shown in Figure 11 (the doubler was in the on state when the switch was closed and in the off state when the switch was open). The measured conversion loss of the doubler for the switch in the on and off configurations is shown in Figure 13a. The obtained results were similar to the ones obtained with the MOSFET switch (see Figure 6b), confirming the correct operation of the doubler.

Among other things, this solution can be used to monitor the rotational speed of machines and motors and can be used for passive wireless sensing just connecting the RF ports of the doubler to two antennas (one operating at f0 and the other at 2f0). In the experiment shown in Figure 14, the doubler with the reed switch was used to measure the rotational speed of an electrical motor. A roll was connected to the motor. Four magnets were applied to the border of the roll, placed at equal distance. Four magnets were used instead of one to increase the measured frequency (the limited resolution bandwidth of the spectrum analyzer used for the acquisition hindered the detection of modulation frequencies below 20 Hz). The reed switch connected to the doubler was placed close to the border of the roll. This way, the reed switch was switched on every time one of the magnets passed in front of it. The measured spectrum is shown in Figure 13b. Although the rotating frequency was close to the limit of the resolution bandwidth of the used spectrum analyzer, we can still see the two side bands occurring at ±40 Hz with respect to the $2f_{0}$ output carrier frequency. The correctness of the measured rotational speed was confirmed also by a photoelectric rotation sensor (that exactly counted the number of turns per second) connected to an oscilloscope.

This experiment highlights the potential of the proposed solution, since the proposed sensing approach cannot be used in the other modulated frequency doublers found in the SoA, due to the large parasitics of the reed switch.

## 5. Discussion

Table 1 illustrates a comparison of the proposed circuit with SoA frequency multipliers. It was shown that the proposed solution could reach a similar performance to the SoA, although it was based on a low-frequency MOSFET switch. Indeed, the proposed solution relied on a different modulation principle, which could lead to a completely passive system. For instance, the MOSFET could be replaced by an electromechanical switch or a variable resistance, which do not require any electrical signals for their activation (see Section 4). Moreover, the proposed system operated with an $f_{m}$ higher than the other modulation-capable frequency doublers working at similar RF frequencies.

Comparing [22,23,24] with the approach described in this paper, one can notice three main differences. First, the cited designs all refer to mm-wave and sub-THz telecom systems, whereas our circuit is devoted to sensing applications in the UHF to low-GHz frequency range. Secondly, the above frequency multipliers are active, while that presented here is passive. Third, the modulation mechanisms are completely different. The sub-THz multipliers are modulated either by means of transistors acting as RF switches or switching on and off the bias of the circuit itself (amplifier or multiplier stages). In this paper, instead, the modulation is achieved varying the dc termination impedance experienced by the Schottky diode, a method that makes the bias not (strictly) necessary, as shown in Section 4.

## 6. Conclusions

A novel passive frequency doubler, enabling the amplitude modulation of its second-harmonic output signal by changing the dc impedance termination of a diode, was presented. The schematic diagram and operating principle of the circuit were discussed. Since the amplitude modulation was achieved by applying a periodic signal to the gate port of a low-frequency MOSFET (called modulation port), no static power consumption was experienced. A prototype in microstrip technology was manufactured and tested. A variation of 23 dB was observed in the conversion loss of the circuit when the MOSFET was on and off. Finally, a square wave was applied to the modulation port ($f_{m}$ up to 15 MHz), and the measured spectrum of the output signal showed a good agreement with the developed theory. Finally, the MOSFET was replaced by a reed switch and the capability of the doubler to measure the rotational speed of a motor without using any active circuit was demonstrated. The results open the door to a new class of frequency doublers, suitable for ultra low-power harmonic transponders for identification and sensing applications.

## Figures and Tables

**Figure 1 sensors-22-00620-f001:**
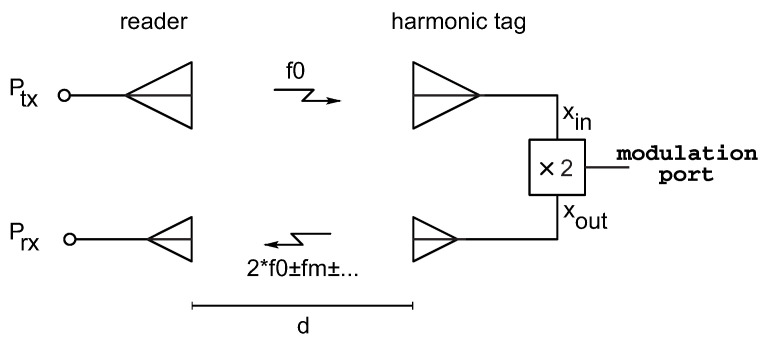
Block diagram of the proposed system, based on a modulation-capable Schottky-diode frequency doubler.

**Figure 2 sensors-22-00620-f002:**
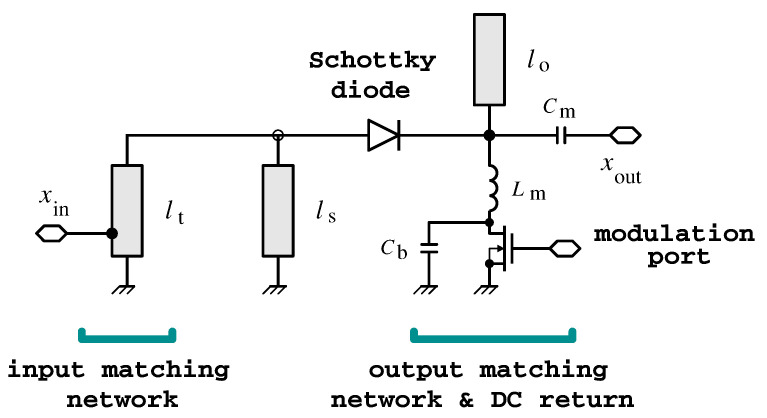
Schematic diagram of the frequency doubler with a modulation port. Main parameters: $l_{t}=43$ mm, $l_{s}=50.5$ mm, $l_{o}=41$ mm, $L_{m}=6$ nH, and $C_{m}=1$ pF. A BSS123 (ON-Semiconductors) MOS transistor is used to perform the OOK modulation.

**Figure 3 sensors-22-00620-f003:**
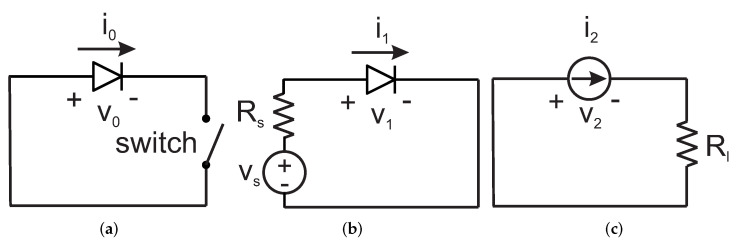
Equivalent circuits for the different signal components flowing through the doubler: (**a**) dc component, (**b**) fundamental frequency $f_{0}$, and (**c**) second harmonic $2f_{0}$.

**Figure 4 sensors-22-00620-f004:**
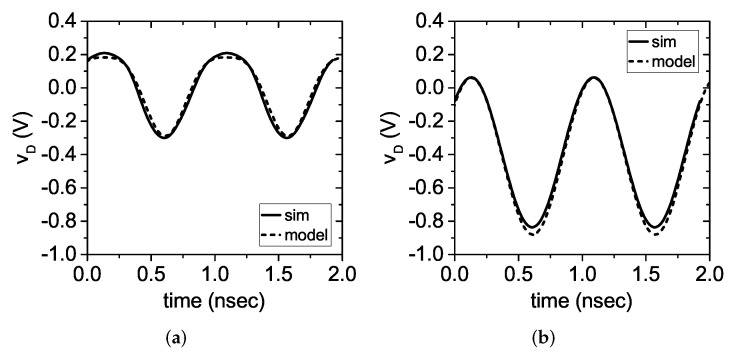
Diode voltage $v_{D}$ in time domain at $P_{in}=-10$ dBm: comparison between simulation and theoretical model (**a**) with the MOSFET switch closed (on) and (**b**) with the MOSFET switch open (off).

**Figure 5 sensors-22-00620-f005:**
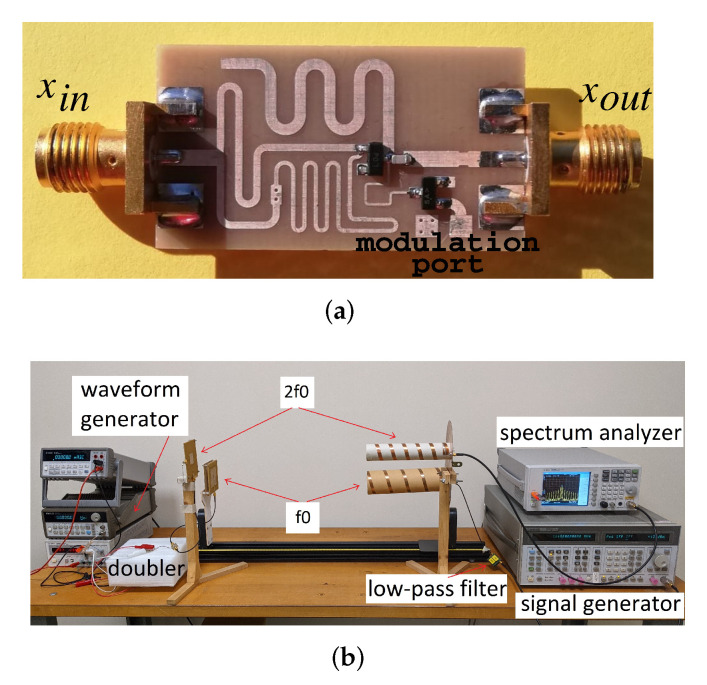
Harmonic transponder for both identification and sensing. (**a**) photo of the frequency doubler prototype with modulation port. (**b**) photo of the wireless experimental setup. Frequency doubler PCB area: $15\times 23\;{\mathrm{mm}}^{2}$.

**Figure 6 sensors-22-00620-f006:**
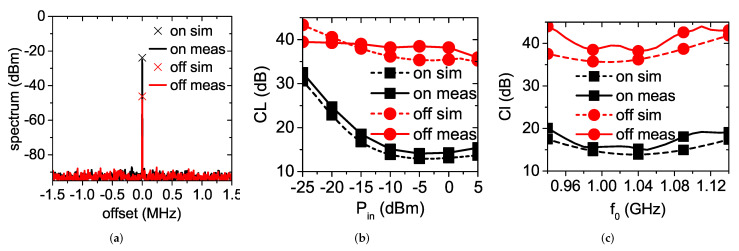
Doubler performance in static conditions. (**a**) output spectrum when the MOSFET is on and off ($f_{0}=1.04$ GHz and $P_{in}=-10$ dBm). Conversion loss of the doubler when the MOSFET is on and off (**b**) versus $P_{in}$ ($f_{0}=1.04$ GHz) and (**c**) versus $f_{0}$ ($P_{in}=-10$ dBm).

**Figure 7 sensors-22-00620-f007:**
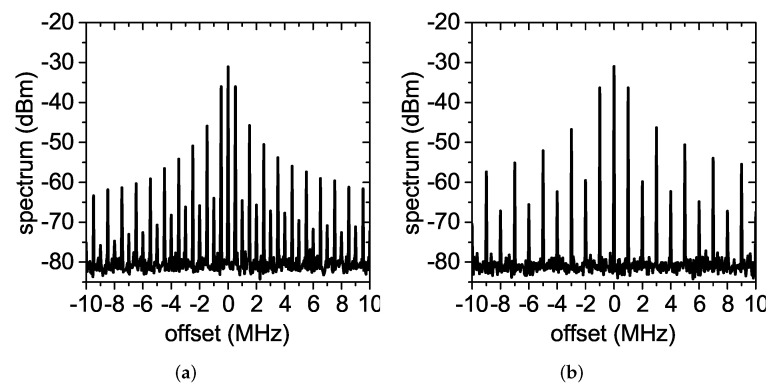
Output spectrum of the modulation-capable frequency doubler with $P_{in}=-10$ dBm. (**a**) $f_{m}=500$ kHz, (**b**) $f_{m}=1$ MHz, (**c**) $f_{m}=10$ MHz, and (**d**) $f_{m}=15$ MHz.

**Figure 8 sensors-22-00620-f008:**
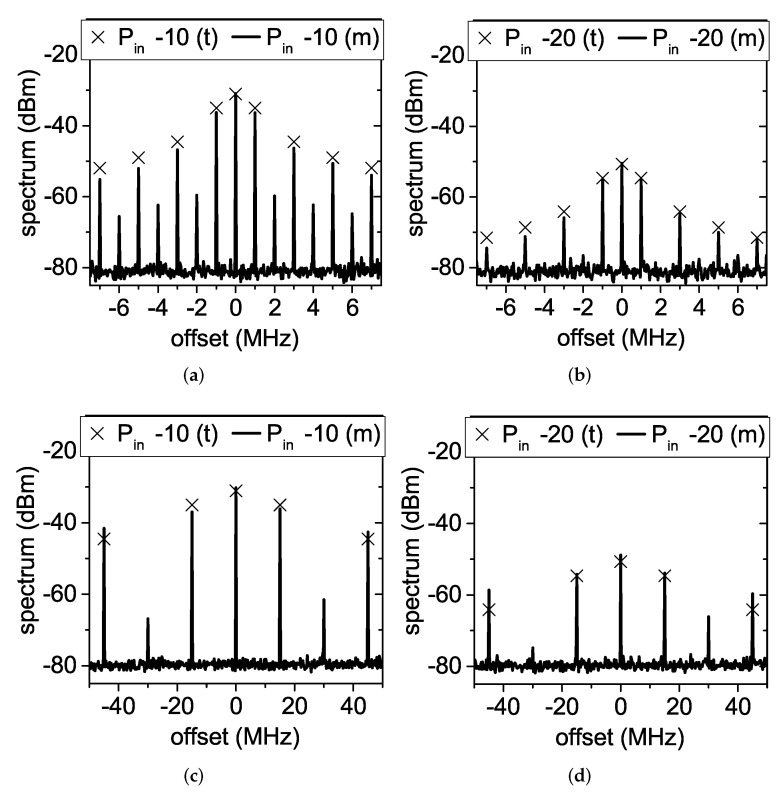
Output spectrum of the modulation-capable frequency doubler: comparison between theory and measurements. (**a**,**b**) $f_{m}=1$ MHz and (**c**,**d**) $f_{m}=15$ MHz. (**a**,**c**) $P_{in}=-10$ dBm and (**b**,**d**) $P_{in}=-20$ dBm. “t” stands for theory and “m” for measurements.

**Figure 9 sensors-22-00620-f009:**
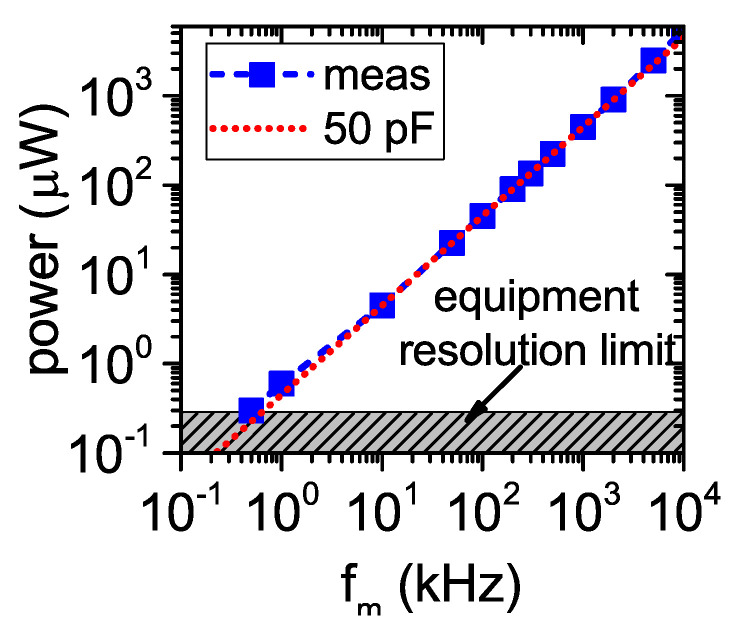
Dynamic power consumption of the modulator as a function of the modulation frequency. The equivalent capacitor C=50 pF is estimated from $C=P_{c}/\left(V_{on}^{2}f_{m}\right)$, where $V_{on}=3$ V, with the aim to obtain the best match with the measurement results in the modulation frequency range of interest.

**Figure 10 sensors-22-00620-f010:**
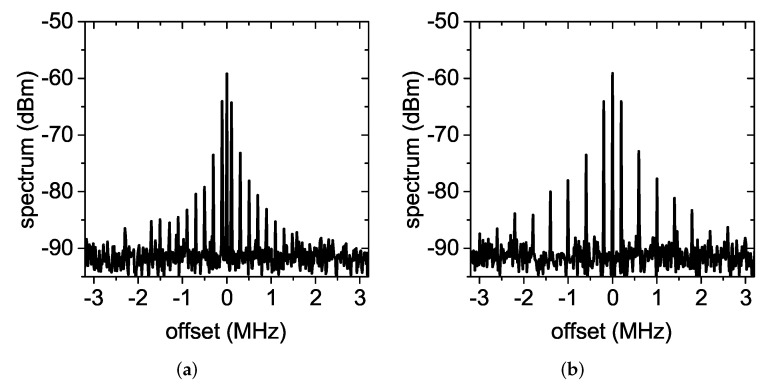
Spectrum received from the complete wireless harmonic transponder for diverse modulation frequencies. (**a**) $f_{m}=100$ kHz, (**b**) $f_{m}=200$ kHz, (**c**) $f_{m}=500$ kHz, and (**d**) $f_{m}=1$ MHz.

**Figure 11 sensors-22-00620-f011:**
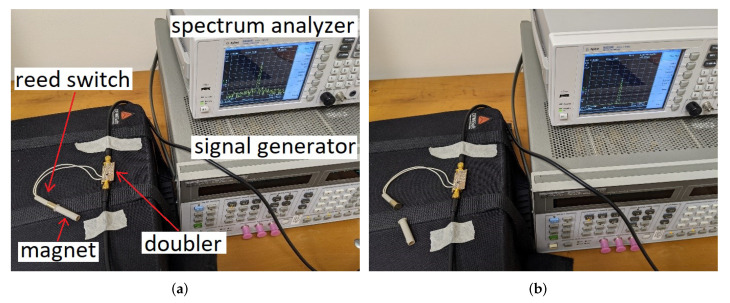
Frequency doubler modulated by means of a reed switch. (**a**) reed switch on. (**b**) reed switch off.

**Figure 12 sensors-22-00620-f012:**
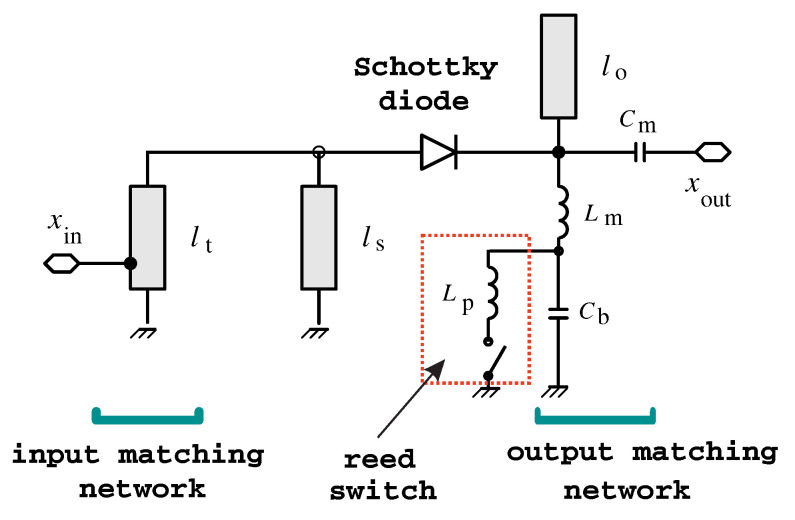
Schematic diagram of the frequency doubler modulated by a reed switch. A low-frequency reed switch for industrial application is used in the experiments. Lp models the parasitic switch inductance, including the connecting wires, and is on the order of μH.

**Figure 13 sensors-22-00620-f013:**
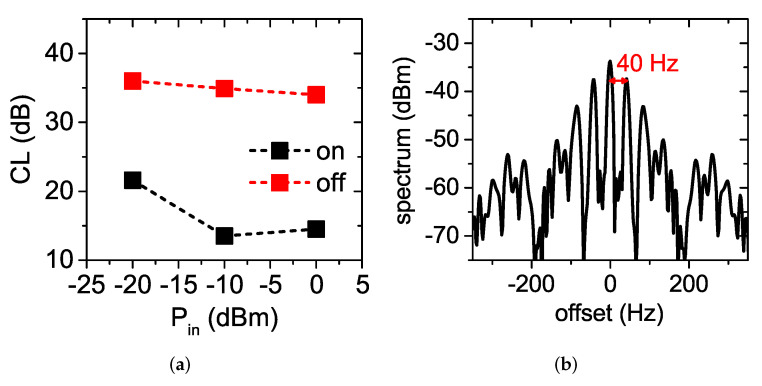
Measured results of the frequency doubler modulated by a reed switch. (**a**) Conversion loss of the doubler with the reed switch in on and of configuration. (**b**) Output spectrum of the doubler in the dynamic experiment.

**Figure 14 sensors-22-00620-f014:**
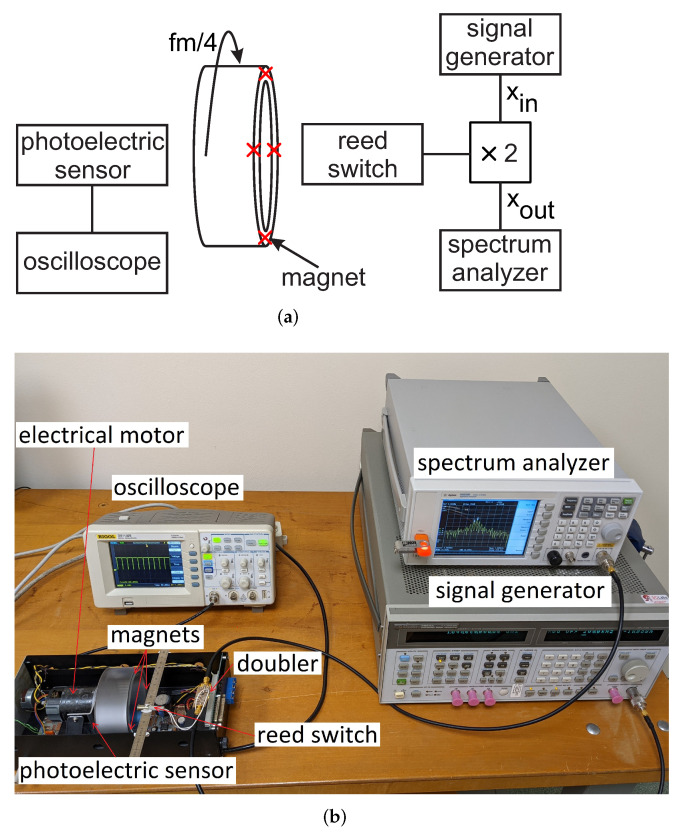
Frequency doubler modulated by a reed switch, used to monitor the rotational speed of an electric motor: (**a**) block diagram, and (**b**) photo of the experimental setup.

**Table 1 sensors-22-00620-t001:** Comparison with SoA modulation-capable frequency multipliers.

Ref.	Mod. Principle	Tech.	Multiplier Type	$P_{\mathrm{in}}$ (dBm)	CL (dB)	On/Off (dB)	${\mathrm{nf}}_{0}$ (GHz)	$f_{m}$ (MHz)
this work	dc term. variation	Schottky	passive	−10	15.1	−23	2.08	15
[13]	bias variation	Schottky	passive	/	/	/	1.736	0.032
[19]	bias variation	varactor	passive	−10	15	26	868	0.0328
[20]	bias variation	varactor	passive	/	/	/	4.4	0.005
[21]	input match	Schottky	passive	0	23	44	1.83	0.15
[22]	out amp on/off	InP HBT	active	3	−2	20	240	14,000
[23]	driver amp on/off	SiGe HBT	active	0	−6.5	20	324	7500
[24]	outphasing	SiGe HBT	active	/	/	33	216	8000

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
