# Peer review of "Energy-Efficient Harmonic Transponder Based on On-Off Keying Modulation for Both Identification and Sensing"

_sensors, 2022, doi:10.3390/s22020620_

Round 1

Reviewer 1 Report

I thank the authors for writing up this work about a harmonic transponder for on-off keying (OOK) modulation in backscatter applications. The paper presents a low-power circuit topology that enables amplitude modulation of a harmonic transponder. The paper presents a clear story, and I appreciate the design, simulation and measurements provided by the authors.

In my opinion, the downside to the paper is that it does not present sufficient novelty or detail. Most of the design and analysis appears to come from the the authors' previous work in [24]. This paper does present an elegant method for adding OOK modulation, however, the analysis is lacking useful information beyond the circuit topology that is critical for readers. For example, the authors state that the benefit of this application is the ability to use low-frequency switching mechanisms, but there is little justification as to why this is so. Additionally, it would be quite helpful to have more analysis as to why the modulation signal is limited to fm <= 1 MHz. This information would provide readers a better idea of novelty of this work in the realm of RF electronics, more so than just seeing that a switching element was added to a previous design.

Specific comments are included below:

  • The paper would benefit from a high-level block diagram figure at the beginning that describes the operation of the circuit and provides context for Fig. 1.
  • No mention in the paper about the metric "Cg" that is used in Table 1.
  • Lines 55-59: Last paragraph of the intro  could more clearly state the novelty of the work relative to other publications.
  • Line 62: the abbreviation RF for radiofrequency has already been used.
  • Line 66: slightly more explanation (1-2 sentences) about how the stubs are operation would make the circuit clearer.
  • Line 79: "that" should be "which"
  • Lines 87: "reversely" should be "reverse"
  • Line 92: Suggest writing "Cl" as "CL", since the lower-case "L" is easily confused with an upper-case "i"
  • Line 96: "only on the dc signal"--"on" should be removed
  • Line 106: why did you choose the frequency 1.04 GHz? Seems like a arbitrary choice. Why not use a frequency in the ISM band?
  • Fig. 3: could you first include  a plot of the spectrum with the MOSFET on and off at Pin = -10 dBm and at fo = 1.04 GHz? This would be useful for putting the other figures in context.
  • Fig. 3(c): seems out of place. May be more relevant in Fig. 4 when discussing the impact of different modulation frequencies.
  • Fig. 3(c): Is the resolution limit based on your measurement equipment? If so, please state "equipment resolution limit" for clarity.
  • Line 151: Please use a peer-reviewed source for referencing the dynamic power consumption of the modulator. Reference 26 seems to be a poster for a competition.
  • Line 153: Did you determine the input capacitance of the MOSFET? Does the estimate of 50pF match this?
  • Fig. 4: It is very hard to analyze the plots when the paper is in grayscale. Consider making the spectra easier to differentiate.
  • Fig. 4(b): What is the origin of the harmonic at 2 MHz offset? Was this not found in the theory?
  • Fig. 4(b): x-axis limits should be extended, the edge of the plot is cutoff.
  • Fig. 4(b): What do you think accounts for the discrepancy between measurement and theory as the offset frequency increases?
  • Lines 162-179 and Table 1: These seem to be more appropriate as part of the Discussion or Conclusion sections.
  • Line 171: typo for mm-wave
  • Lines 170-179: The works from Table 1 are summarized, but the comparison seems somewhat superficial. Are there only works in the mm-wave and sub-THz regimes to which you can compare? For example, Schwerdt HN, Xu W, Shekhar S, et al. A Fully-Passive Wireless Microsystem for Recording of Neuropotentials using RF Backscattering Methods. J Microelectromech Syst. 2011;20(5):1119-1130. doi:10.1109/JMEMS.2011.2162487
  • The experiment with the motor seems better suited as a sub-section of Experimental Results rather than being the Discussion section.

Reviewer 2 Report

The paper purpose simple harmonic transponders for both identification and sensing application based on Schottky-diode frequency doubler associated with on-off keying (OOK) modulation port. the paper is well written and supported with an adequate set of experiments including a sensing scenario(strong point). 

Some points should be addressed:

-It is good to mention some limitations of harmonic transponders including the conversion loss and higher path loss at 2f.

-The authors claim that the frequency doubler operates as in [24] have a moderate conversion loss. .the loss greater than 12 dB  is not moderate(maybe in comparison to referenced works)

-eq 1 considers that the output signal is the multiplication of the second harmonic generated by the diode by a square wave.  I think is more adequate to write the polynomial behavior of the diode and exclude only the component at f0 (with the oped stub )

-The choice of the diode should be justified
- The tag-to-reader distance is 50 cm is in the near field. please comment 

- It is important to comment on the maximum possible range with the tag  sensitivity

- Figure 6. with Frequency doubler modulated by mean of a reed switch is not clear at all. please improve the quality or avoid this setup photo
- Fig 8, it will be good to add a building block illustration.
